# Scattered Radiation Distribution Utilizing Three Different Cone-Beam Computed Tomography Devices for Maxillofacial Diagnostics: A Research Study

**DOI:** 10.3390/jcm12196199

**Published:** 2023-09-26

**Authors:** Sotirios Petsaros, Emmanouil Chatzipetros, Catherine Donta, Pantelis Karaiskos, Argiro Boziari, Evangelos Papadakis, Christos Angelopoulos

**Affiliations:** 1Department of Oral Diagnosis and Radiology, Faculty of Dentistry, National and Kapodistrian University of Athens, 2 Thivon Street, Goudi, 11527 Athens, Greece; sotirios100@gmail.com (S.P.); e.chatzipetros@gmail.com (E.C.); edonta@dent.uoa.gr (C.D.); epapadak@dent.uoa.gr (E.P.); 2Medical Physics Laboratory, Faculty of Medicine, National and Kapodistrian University of Athens, 75 Mikras Asias Street, Goudi, 11527 Athens, Greece; pkaraisk@med.uoa.gr; 3Greek Atomic Energy Commission, Agia Paraskevi, 15310 Attiki, Greece; argiro.boziari@eeae.gr

**Keywords:** cone-beam computed tomography, dosimetry, radiation protection, scattered radiation

## Abstract

This study aimed to estimate scattered radiation and its spatial distribution around three cone-beam computed tomography (CBCT) devices, in order to determine potential positions for an operator to stand if they needed to be inside the CBCT room. The following devices were tested: Morita Accuitomo (CBCT1), Newtom Giano HR (CBCT2), Newtom VGi (CBCT3). Scattered radiation measurements were performed using different kVp, mA, and Field of View (FOV) options. An anthropomorphic phantom (NATHANIA) was placed inside the X-ray gantry to simulate clinical conditions. Scattered measurements were taken with the Inovision model 451P Victoreen ionization chamber once placed at fixed distances from each irradiation isocenter, away from the primary beam. A statistically significant (*p* < 0.001) difference was found in the mean value of the scattered radiation estimations between the CBCT devices. Scattered radiation was reduced with a different rate for each CBCT device as distance was increased. For CBCT1 the reduction was 0.047 μGy, for CBCT2 it was 0.036 μGy, and for CBCT3 it was 0.079 μGy, for every one meter from the X-ray gantry. Therefore, at certain distances from the central X-ray, the scattered radiation was below the critical level of 1 mGy, which is defined by the radiation protection guidelines as the exposure radiation limit of the general population. Consequently, an operator could stay inside the room accompanying the patient being scanned, if necessary.

## 1. Introduction

Cone-beam computed tomography (CBCT) is one of the most important technological achievements in oral and maxillofacial radiology in the last forty years. In time, this has found numerous applications, from diagnostic applications to pre-implant assessment and surgical guidance using specialized software [1]. The main parameter that determines CBCT image quality is image resolution, which refers to the overall detail of the acquired image and is described by the maximum frequency that can be perceived [2]. Resolution is distinguished between spatial resolution and contrast resolution. Spatial resolution is a key intrinsic parameter that characterizes imaging systems and is widely used for their evaluation. It expresses the ability of the imaging system (in mm) to distinguish between two small objects that are very close to each other, in a high-contrast environment, and for this reason it is also called high-contrast discrimination ability [3,4]. Contrast resolution is the parameter that describes the ability of a system to distinguish between small differences in the intensity of the recorded signal and to be able to image anatomical structures with approximately linear attenuation coefficients. Factors affecting resolution are mA, kV, Field of View (FOV), and image reconstruction algorithms. Moreover, general image degradation factors such as noise, radiation scatter, and artifacts may compromise resolution [5].

The X-ray beam of the CBCT machine consists of primary radiation, which yields useful imaging information through the patient, and secondary radiation which is scattered radiation [1]. The primary radiation is produced within the X-ray tube, enters the patient, interacts with human tissues, and attenuates variably in the area under examination, conveying the useful information about the structures to be imaged [1,2]. Scattered radiation is a secondary radiation generated during the interaction of the primary beam with the patient tissues [6]. The scattered photons are of a lower energy and show an altered direction in comparison with that of the primary beam. Thus, scattered radiation has a negative effect on image quality [7] and essentially stands as the main factor contributing to reduced spatial resolution, reduced contrast resolution, and increased noise in CBCT [8,9,10].

The health risks associated with occupational radiation exposure are either of a deterministic or stochastic nature [11]. Stochastic effects occur by chance and include cancer risk. The stochastic effect risk is considered to increase with dose according to the linear-no-threshold model. The International Commission on Radiological Protection has recommended an annual occupational exposure limit of 20 mSv/year, averaged over 5 years, in both effective dose and equivalent eye lens dose [12,13]. These effects can develop independently of the radiation dose, and no threshold effect can be defined. Therefore, added exposures of the patient increase the chance of occurrence of a stochastic effect [14]. Although radiation doses are low during dental practice, there is always a concern in the dental community about radiation exposure [15,16,17,18]. Deterministic effects are limited to a certain threshold dose, and are thus unlikely to appear with a range of dental examination exposures [11].

Our hypothesis assumes that in some exceptional occasions the dentist or the staff (dental assistant, radiology technologist) may need to be present in the X-ray room during the CBCT examination, to visually and verbally direct and encourage the patient to place themselves correctly. Thus, we wanted to determine if there is a safe distance from the CBCT device to stand, so to receive the lowest possible scattered radiation [6,11].

This original research study specifically aims to estimate the patterns of scattered radiation and its spatial distribution around three CBCT devices, in order to determine potential positions for an operator to stand if they need to be present in the X-ray room during the CBCT examination for maxillofacial diagnostics.

## 2. Materials and Methods

### 2.1. Study Material

The following devices were tested in this research study: Morita Accuitomo (CBCT1) (J. Morita Corp., Osaka, Japan), Newtom Giano HR (CBCT2) (Cefla s.c., Bologna, Italy), and Newtom VGi (CBCT3) (Cefla s.c., Bologna, Italy). Exposure measurements were performed for different kVp, mA, and Field of View (FOV) values ((CBCT1; exposure time: 9 s, kVp: 90 kV, mA: 7 mA, voxel size: 0.125 mm), (CBCT2; exposure time: 3.6 s, kVp: 90 kV, mA: 3.66 mA, voxel size: 0.125 mm), (CBCT3; exposure time: 3.6 s, kVp: 110 kV, mA: 3.66 mA, voxel size: 0.3 mm)). An anthropomorphic phantom (NATHANIA) (Computerized Imaging Reference Systems, CIRS, Inc., Norfolk, VA, USA) was placed in the X-ray gantry to imitate clinical conditions (Figure 1).

In terms of ambient dose equivalent H*(10), scattered radiation measurements were taken with a Victoreen ionization chamber (Inovision 451P), with dimensions 10 cm × 20 cm × 15 cm (451P) (Fluke Biomedical Radiation Management Services, Cleveland, Ohio, USA). Calibration of ionization chambers provides traceability to Physikalisch-Technische Bundesanstalt (PTB) through the Ionizing Radiation Calibration Laboratory of the Greek Atomic Energy Commission (Secondary Standard Dosimetry Laboratory, (SSDL)). The survey meter was placed at fixed distances from each irradiation isocenter, away from the primary beam [11,19]. The ionization chamber (451P) exhibited a direct response for measuring scattered radiation dose and showed a high detection capability for very small radiation doses, including radiation present in the natural environment and stable energy dependence in the 40–100 keV range. Scattered radiation measurements were performed at the same distances from the CBCT devices (Figure 2).

The placement positions of 451P were determined in each CBCT room based on point “0”, which represented the fixed position of each CBCT device [20]. More specifically, two reference axes were drawn in the floor: the first (*x*-axis) represented the distance (in m) of the 451P to the right and left of the CBCT device, showing positive (right) and negative (left) values, respectively. The second (*y*-axis) represented the distance (in m) of the 451P from the CBCT device, in front of the CBCT device, perpendicular to the *x*-axis (Figure 2) [20].

A total of 191 (CBCT1) and 32 (CBCT2) measurements of scattered radiation at two different heights (1 m or 1.3 m from the floor) were carried out in the rooms of CBCT1 and CBCT2 devices at each point of intersection of the *x* and *y* axes (yellow points) (Figure 2). The measurement at the height of 1 m from the floor represented the anatomical location of the gonads and the height of 1.3 m from the floor represented the anatomical location of the thyroid gland [21]. It is of importance that in the CBCT3 room, all measurements (36 measurements) were carried out at the same height (1.3 m) due to technical difficulties (room and device restrictions).

Measurements of scattered radiation from different Fields of View (FOVs) were also performed. More specifically, measurements of scattered radiation were carried out in the CBCT1 device in three different FOVs ((4 × 4), (6 × 6), (8 × 8)), in the CBCT2 device in two different FOVs ((8 × 8), (11 × 8)), and in the CBCT3 device in two different FOVs ((8 × 8), (15 × 15)).

### 2.2. Statistical Analysis

Data were described using mean values and standard deviation (SD) for the scattered radiation dose measurements (µGy) in the ionization chamber (451P) at three different rooms of CBCT devices (CBCT1, CBCT2, CBCT3). One-way analysis of variance (ANOVA) and independent samples *t*-test were used to assess the mean difference of 451P measurements between the three rooms and between the FOVs in each room. Then, Bonferroni tests for multiple comparison corrections were applied. In order to investigate whether the position of CBCT device, in the three rooms, was related to 451P, the Euclidean distance of each measurement point was calculated as the distance in meters (m). Generalized additive models (GAMs) were applied to assess the relationship between 451P (dependent variable) and distance (m) from the CBCT device (independent variable), in each room. Mixed effect linear regression models were used to assess the relationship between 451P and the FOV, in each room. All models were adjusted for the height (m) that the measurement was carried out and for the coordinates (*x*, *y*) of each measurement point, by including a bivariate smooth function (thin plate spline) of (*x*, *y*). The spatial distribution of scattered radiation in 451P was estimated through the rigging universal interpolation method. A test for trend was applied to investigate the trend of 451P in the FOVs of each room.

All statistical analysis was performed using R version 4.1.3 (10 March 2022), library (mgcv) and library (lme4). All spatial analysis was conducted using ArcGIS Desktop v.10.1. (Spatial Analyst Tools, Interpolation, Spline). Two-tailed *p*-values are reported. A *p*-value less than 0.05 was considered as statistically significant.

## 3. Results

Table 1 presents the number of points and measurements performed in each room, the height of the measurements performed, the distribution of 451P measurements, and the maximum distance value from point “0”. It is worth noting that all measurements were carried out at the same height in room 3. A statistically significant difference was observed in mean 451P measurements (*p* < 0.001) between rooms. Specifically, after applying the Bonferroni test, differences in the mean 451P measurements were statistically significant between rooms 2 and 3 (*p* <0.001), rooms 1 and 3 (*p* <0.001), and between rooms 1 and 2 (*p* < 0.001). The maximum value of 451P measurements (9.03 µGy) was observed in CBCT1, in a distance of 100 cm from point “0”, while in CBCT2 and CBCT3 the maximum values were 5.70 and 8.70 µGy at a distance of 50 cm and 55 cm from point “0”, respectively (Table 1).

Table 2 presents the distribution of 451P measurements, for different FOVs in each room. A statistically significant difference in 451P measurements according to FOV was found in room 1 (*p* = 0.012) and in room 3 (*p* = 0.001). Regarding room 1, a significant difference in 451P measurements was observed between FOVs 4 × 4 and 8 × 8 (Bonferroni multiple comparison *p* = 0.010). Moreover, as the FOV increased, a significant increasing trend in 451P measurements was shown in rooms 1 and 3 (*p* < 0.001). For example, in room 1, 451P measurements made at FOV 6 × 6, compared to FOV 4 × 4, were on average higher by 0.523 μGy (95% Confidence Interval (C.I.): 0.139 to 0.820) μGy). Moreover, 451P measurements performed at FOV 8 × 8, compared to 4 × 4, were on average higher by 0.776 μGy (95% C.I.: 0.365 to 1.053) μGy) (Table 2).

Table 3 shows the results from applying generalized additive models, with 451P measurement as the dependent variable and distance of measurements as the independent variable, also adjusting for height of measurements made (only in rooms 1 and 2) and coordinates (*x*, *y*) of measurement points, in each room. In room 1, a 1 m increase in the distance from the CBCT device, resulted in a decrease in mean 451P by 0.047 μGy (95% C.I.: −0.057 to −0.037 μGy), adjusting for the other variables. In room 2, a 1 m increase in the distance from the CBCT device, resulted in a decrease in mean 451P by 0.036 μGy (95% C.I.: −0.062 to −0.010 μGy), taking into account the other variables. Note that in both rooms, the height of the measurements did not significantly predict 451P (*p* = 0.956 and *p* = 0.323, respectively). In room 3, a 1 m increase in the distance from the CBCT device resulted in a decrease in mean 451P by 0.079 μGy (95% C.I.: −0.115 to −0.043 μGy), adjusting for the other variables (Table 3).

Table 4 presents the results from linear mixed effect regression models, with 451P measurement as the dependent variable and FOV as the independent variable, also adjusting for height of measurements made (only for room 1 and 2) and coordinates (*x*, *y*) of measurement points, in each room. A statistically significant effect between FOV and 451P was found in rooms 1 and 3. Specifically in room 1, FOV 6 × 6 and 8 × 8, compared to FOV 4 × 4, had on average a higher value of 451P by 0.480 (95% C.I.: 0.139 to 0.820 μGy) and 0.709 μGy (95% C.I.: 0.365 to 1.053 μGy), respectively. In room 3, FOV 15 × 15, compared to FOV 8 × 8, had a higher mean value of 451P by 2.005 μGy (95% C.I.: 1.453 to 2.558 μGy) (Table 4).

The spatial distribution of scattered radiation in 451P was estimated through the rigging universal interpolation method. Color maps of dose distributions were drawn for horizontal and vertical planes. Scattered radiation dose mapping in the ionization chamber (451P) was depicted per room (CBCT1, CBCT2, CBCT3) on the scattered radiation dose distributions color maps (Figure 3, Figure 4 and Figure 5). It is worth noting that color maps in rooms 1 and 2 appeared to be uniform (CBCT1, CBCT2), regardless of the measurement height. Measurement height did not statistically significantly differentiate the measurement of scattered radiation in the ionization chamber.

**Figure 3 jcm-12-06199-f003:**
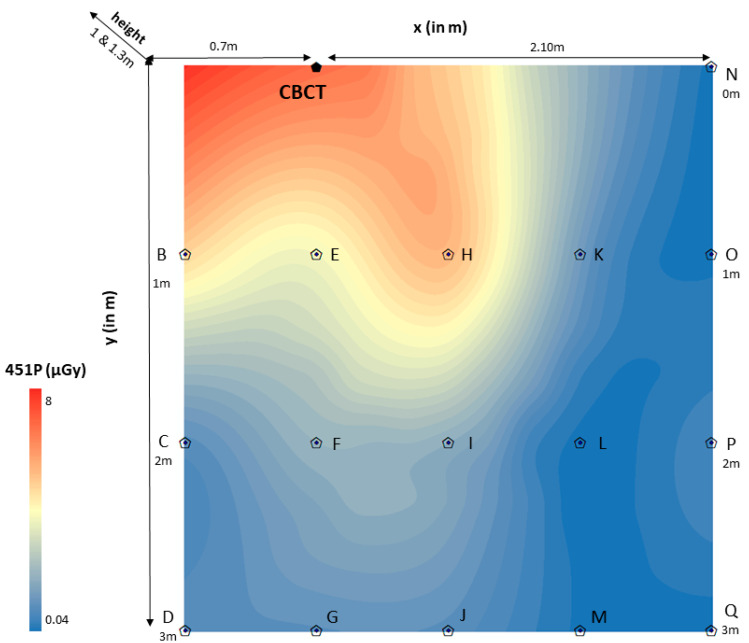
Color map of spatial distribution of scattered radiation (μGy) in room 1 (2.8 × 3 m).

CBCT: Cone-beam computed tomography device 1 (CBCT1); point “0”.x (in m): *x*-axis defined the distance (in m) of the ionization chamber (451P) to the right and left of the CBCT1 device (point “0”), showing sometimes positive (right) and sometimes negative (left) values. In this color map, the absolute values are displayed in order to avoid confusion ((point A − point “0” = 0.7 m), (point “0” − point N = 2.10 m)).y (in m): *y*-axis defined the distance (in m) of the ionization chamber (451P) from the CBCT1 device in an anterior position, perpendicular to the *x*-axis ((point A − point D = 3 m), (point N − point Q = 3 m)).451P (μGy): Scattered radiation measurements (μGy) were taken with the Inovision model 451P Victoreen ionization chamber (0.04–8μGy). Red color means very high scattered radiation dose, while blue color means very low scattered radiation dose.Height (1 and 1.3 m): The measurement carried out at a height of 1 m from the floor represented the anatomical location of the gonads. The measurement carried out at a height of 1.3 m from the floor represented the anatomical location of the thyroid gland.

**Figure 4 jcm-12-06199-f004:**
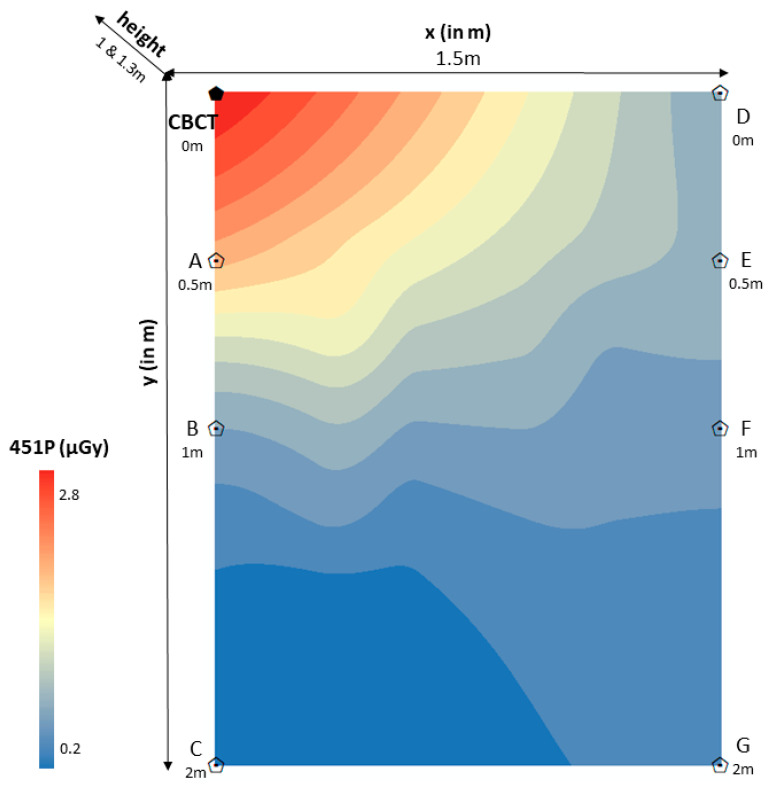
Color map of spatial distribution of scattered radiation (μGy) in room 2 (1.5 × 2 m).

CBCT: Cone-beam computed tomography device 2 (CBCT2); point “0”.x (in m): *x*-axis defined the distance (in m) of the ionization chamber (451P) to the right of the CBCT2 device (point “0”), showing positive values (point “0” − point D = 1.5 m).y (in m): *y*-axis defined the distance (in m) of the ionization chamber (451P) from the CBCT2 device in an anterior position, perpendicular to the *x*-axis ((point “0” − point C = 2 m), (point D − point G = 2 m)).451P (μGy): Scattered radiation measurements (μGy) were taken with the Inovision model 451P Victoreen ionization chamber (0.2–2.8 μGy). Red color means very high scattered radiation dose, while blue color means very low scattered radiation dose.Height (1 and 1.3 m): The measurement carried out at a height of 1 m from the floor represented the anatomical location of the gonads. The measurement carried out at a height of 1.3 m from the floor represented the anatomical location of the thyroid gland.

**Figure 5 jcm-12-06199-f005:**
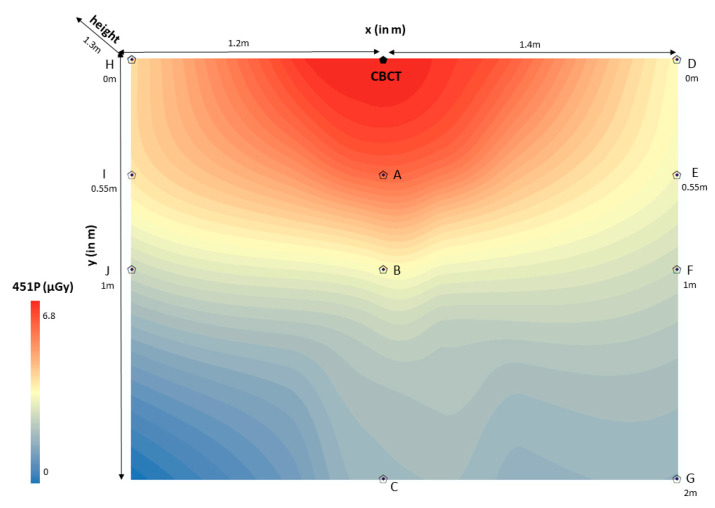
Color map of spatial distribution of scattered radiation (μGy) in room 3 (2 × 2.6 m).

CBCT: Cone-beam computed tomography device 3 (CBCT3); point “0”.x (in m): *x*-axis defined the distance (in m) of the ionization chamber (451P) to the right and left of the CBCT3 device (point “0”), showing sometimes positive (right) and sometimes negative (left) values. In this color map, the absolute values are displayed in order to avoid confusion ((point H − point “0” = 1.2 m), (point “0” − point D = 1.4 m)).y (in m): *y*-axis defined the distance (in m) of the ionization chamber (451P) from the CBCT3 device in an anterior position, perpendicular to the *x*-axis ((point H − point J = 0.55 m), (point D − point G = 2 m)).451P (μGy): Scattered radiation measurements (μGy) were taken with the Inovision model 451P Victoreen ionization chamber (0–6.8μGy). Red color means very high scattered radiation dose, while blue color means very low scattered radiation dose.height (1.3 m): The measurement carried out at a height of 1.3 m from the floor represented the anatomical location of the thyroid gland.

## 4. Discussion

In this research study, measurements of the scattered radiation dose were collected inside rooms with CBCT installations. The spatial distribution of scattered radiation measured with ionization chamber (451P) was estimated through the rigging universal interpolation method and the safest locations of the people who could be present inside the X-ray room were determined (>100 cm from CBCT1, >50 cm from CBCT2, and >55 cm from CBCT3).

The protection against scattered radiation is a perennial concern of the scientific community, even when the radiation dose is quite low, such as during intraoral radiography [20,22,23]. A research study showed that occupationally exposed individuals presented a higher incidence of thyroid cancer, especially in the past when the radiation protection measurements were not as strict [21]. A similar epidemiological study in Canada argued that repeated exposure to low doses by occupation was limited to long-term harmful effects and cancer incidence [24]. Cewe et al. (2022) demonstrated that staff can use freestanding radiation protection shields instead of heavy aprons during intraoperative CBCT imaging, to achieve effective whole body dose reduction with improved comfort [11].

Alcaraz et al. (2006) measured the scattered radiation at various distances from the patient, who was lying supine, 48 cm from the floor. The measurements were carried out at distances of 60, 90, 120, 150, and 180 cm, and at an angle of 0°, 135°, and 180°. The results of this study in relation with dose reduction of intraoral dental radiography showed that the safest position for the dentist was behind and right of the X-ray beam at an angle of 135° [22]. These findings were in agreement with the results of a previous study by Rolofson et al. (1969), who studied radiation isoexposure curves of scattered radiation around a dental chair during radiography. Rolofson et al. (1969) reported that the most appropriate location with the lowest absorbed radiation dose to the gonadal anatomical region was directly behind the X-ray beam or to the side of the patient’s head, opposite the X-ray beam [23]. Yamaji et al. (2021) noticed that if a physician or staff member needs to observe the patient near the table, it would be recommended to stand in the back of the base CBCT device. With the use of a ceiling-mounted transparent lead-acryl screen and a table suspended lead curtain, the doses were reduced 45–92% at a direction of 210° degrees and a distance of 120 cm [19]. In our study, the safest positions of people within the CBCT area were proposed to be >100 cm from the CBCT1 device, >50 cm from the CBCT2 device, and >55 cm from the CBCT3 device. Therefore, our findings were in agreement with previous studies. A limitation of our study was that we did not use angle measurements when placing the ionization chamber (451P) in the three rooms with the CBCT devices.

An increase in the scattered radiation at a height of 100 cm from the floor, at the level of the X-ray gantry, and a decrease in the absorbed dose of radiation near the gantry, were observed by other researchers [6]. Conversely, an increase in scattered radiation behind the gantry was observed during head imaging with computed tomography (General Electric Hi Speed Advantage CT) [25]. Various research studies showed that scattered radiation decreases as we move away from the X-ray beam. More specifically, at a distance of 10 and 20 cm from the X-ray beam in a third-generation computed tomography device, the scattered radiation was detected at high levels of 10 and 18 mSv, while it was greatly reduced, to 2 mSv, at a distance of 30 cm from the X-ray beam [26]. In the present study, it was observed to a statistically significant extent that, in room 1, a 1 m increase in the distance from the CBCT device resulted in a decrease in mean 451P by 0.047 μGy (95% C.I.: −0.057 to −0.037 μGy), adjusting for the other variables. Moreover, in room 2, a 1 m increase in the distance from the CBCT device resulted in a decrease in mean 451P by 0.036 μGy (95% C.I.: −0.062 to −0.010 μGy), taking into account the other variables. Furthermore, in room 3, a 1 m increase in the distance from the CBCT device resulted in a decrease in mean 451P by 0.079 μGy (95% C.I.: −0.115 to −0.043 μGy), adjusting for the other variables. These findings were in agreement with the results of previous studies. However, this by no means implies that our results suggest that radio protection rooms with shielding of the walls are unnecessary.

Yamaji et al. (2021) measured the distribution of scattered radiation by C-arm cone-beam computed tomography (CBCT) in the angiographic suite. In this study, the measurements showed the highest radiation dose over 600 μGy by a single CBCT image acquisition at a distance of 60 cm from the beam entry site and a height of 90 cm from the floor [19]. In the present study, the safest positions of the people who can be found within the CBCT area were proposed to be >100 cm from the CBCT1 device, >50 cm from the CBCT2 device, and >55 cm from the CBCT3 device. However, the values obtained from the measurements were overall much lower than 1 mGy, which is defined by the radiation protection guidelines as the exposure radiation limit of the general population [27]. A comparative advantage of our study was that the measurement of the scattered radiation was carried out in three different CBCT devices and numerous measurements of the scattered radiation were carried out at two different heights. The measurement made at a height of 1 m represented the anatomical region of the gonads. The measurement carried out at a height of 1.3 m represented the anatomical region of the thyroid gland. It was noted, however, that the height of the measurements did not appear to statistically significantly differentiate the measurement of the scattered radiation in the ionization chamber (451P) (*p* > 0.05). In addition, due to technical difficulties, a limitation of the present study was that in room 3 (CBCT3) all measurements were made at the same height (1.3 m). The present study provides future prospects for further investigation of scattered radiation distribution with more CBCT devices for maxillofacial diagnostics.

## 5. Conclusions

In all CBCT devices that were tested in this study, the scattered radiation that an individual may be exposed significantly decreased with distance. The Newtom VGi CBCT showed the greatest decrease with distance. In all instances the measured scattered radiation was below 1 mGy, which is defined by the radiation protection guidelines as the exposure radiation limit of the general population. Nevertheless, the scattered radiation is significantly reduced, as long as the dentist, radiology technologist, or other occupationally exposed individual stands at a safe distance and position from the patient (as determined by our results) in the X-ray room during the CBCT examination for maxillofacial diagnostics.

## Figures and Tables

**Figure 1 jcm-12-06199-f001:**
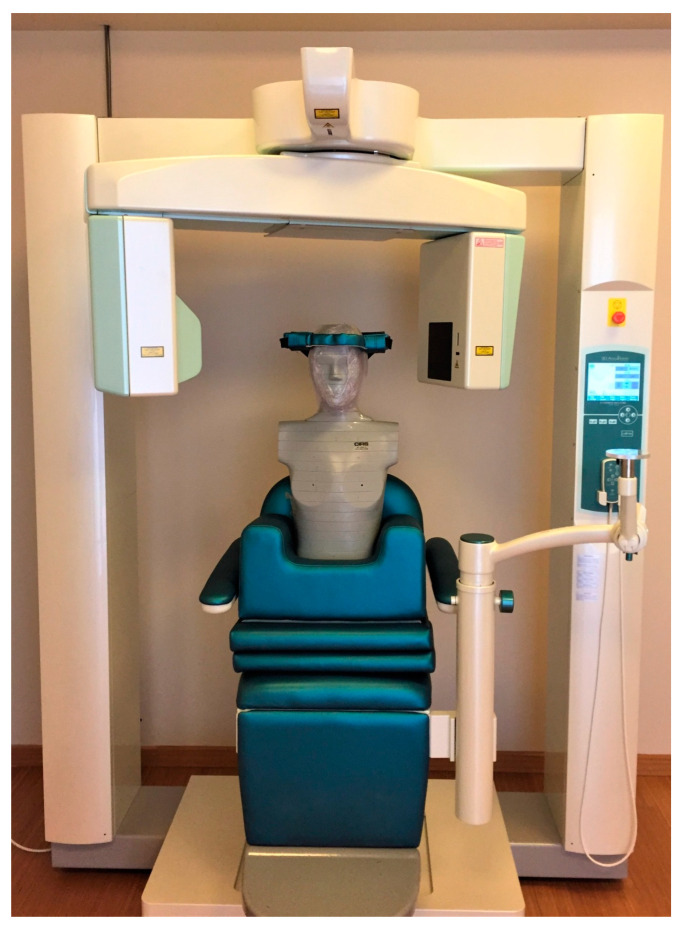
Anthropomorphic phantom (NATHANIA) was placed in the X-ray gantry of the cone-beam computed tomography (CBCT) device to imitate clinical conditions.

**Figure 2 jcm-12-06199-f002:**
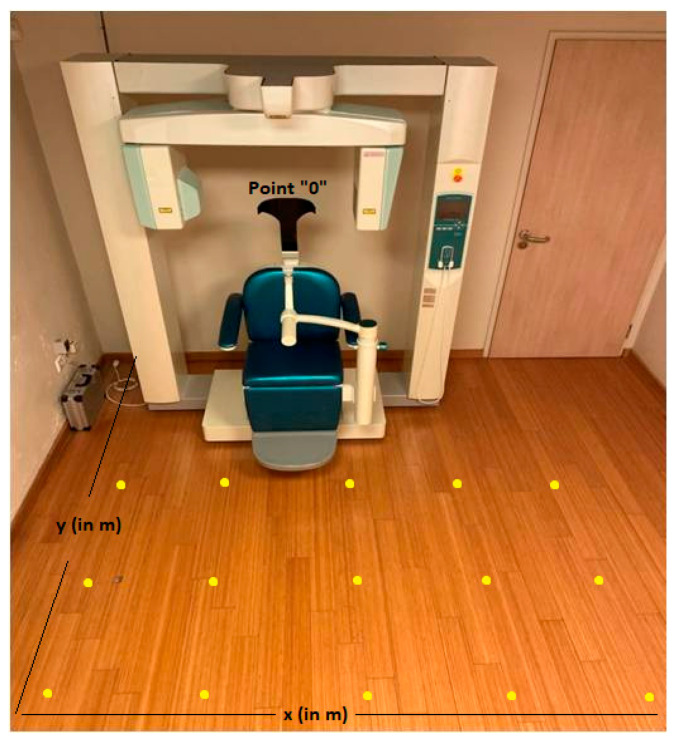
Topographic drawing of the placement positions of the Victoreen ionization chamber, Inovision model 451P (451P). Point “0” represented the fixed position of the cone-beam computed tomography (CBCT) device in each room. The *x*-axis (in m) (thin black horizontal line) represented the distance of the 451P to the right and left from the CBCT device. The *y*-axis (in m) (thin black vertical line) represented the distance of 451P from the CBCT device, in front of the CBCT device, perpendicular to the *x*-axis. The yellow points represented the fixed positions of the 451P at the same distances from the CBCT device (1 m or 1.3 m from the floor).

**Table 1 jcm-12-06199-t001:** Distribution of scattered radiation (451P) (μGy) measurements performed, by room.

			Scattered Radiation 451P Measurements (μGy)	
Room (CBCT)	Points/Measurements (n)	Height (m)	Mean (SD)	Maximum (Distance from point “0”)
1	24/191	1/1.3	1.27 (±1.60)	9.03 (100 cm from point “0”)
2	7/32	1/1.3	0.84 (±1.06)	5.70 (50 cm from point “0”)
3	10/36	1.3	2.86 (±2.21)	8.70 (55 cm from point “0”)

Room (CBCT): X-ray room during cone-beam computed tomography (CBCT). Points: The points represented the fixed positions of the Victoreen ionization chamber (Inovision 451P) at the same distances from the CBCT device; measurements (n): the number of measurements of scattered radiation that were carried out in the rooms (CBCT1, CBCT2, CBCT3). Height (m): The measurement at the height of 1 m from the floor represented the anatomical location of the gonads, and the height of 1.3 m from the floor represented the anatomical location of the thyroid gland. Scattered radiation 451P measurements (μGy): Scattered radiation measurements were taken with a Victoreen ionization chamber (Inovision 451P). Mean: mean value of scattered radiation (451P) (μGy) measurements. SD: standard deviation of scattered radiation (451P) (μGy) measurements. Maximum: maximum value of scattered radiation (451P) (μGy) measurements. Distance from point “0”: Distance (cm) from CBCT device. Point “0” represented the fixed position of the CBCT device in each room.

**Table 2 jcm-12-06199-t002:** Distribution of scattered radiation (451P) (μGy), by Field of View (FOV) and room.

Room (CBCT)	FOV	Scattered Radiation 451P Measurements (μGy) Mean (SD)	*p*-Value	*p*-Value Test for Trend
1	4 × 4	0.652 (±1.017)		
	6 × 6	1.175 (±1.403)	0.012 * ^1^	<0.001 **
	8 × 8	1.428 (±1.696)		
2	8 × 8	0.948 (±1.362)	0.559 ^2^	0.473
	11 × 8	0.723 (±0.581)		
3	8 × 8	1.521 (±0.818)	0.001 * ^2^	<0.001 **
	15 × 15	3.924 (±2.395)		

Room (CBCT): X-ray room during cone-beam computed tomography (CBCT). FOV: Field of View; Scattered radiation 451P measurements (μGy): Scattered radiation measurements were taken with a Victoreen ionization chamber (Inovision 451P). Mean: mean value of scattered radiation (451P) (μGy) measurements. SD: standard deviation of scattered radiation (451P) (μGy) measurements. ^1^ One-way analysis of variance (ANOVA) ^2^ independent samples *t*-test * statistically significant, α = 5% ** statistically significant, α = 1‰.

**Table 3 jcm-12-06199-t003:** Beta coefficient (β) and corresponding 95% Confidence Interval (C.I.) from generalized additive models, with measurements of scattered radiation in ionization chamber (451P) as the dependent variable and distance of measurements as the independent variable, also adjusting for height of measurements ^1^ made and coordinates (*x*, *y*) of measurement points, by room.

Room (CBCT)	Distance (m)		
	β (μGy)	95% C.I. for β (μGy)	*p*-Value
1	−0.047	(−0.057 to −0.037)	<0.001 **
2	−0.036	(−0.062 to −0.010)	0.012 *
3	−0.079	(−0.115 to −0.043)	<0.001 **

^1^ height of measurements varies only in rooms 1 and 2. Room (CBCT): X-ray room during cone-beam computed tomography (CBCT). Distance (m): 1 m increase in the distance (m) from CBCT device. C.I.: Confidence Interval; β: beta coefficient * statistically significant, α = 5% ** statistically significant, α = 1‰.

**Table 4 jcm-12-06199-t004:** Beta coefficient (β) and corresponding 95% Confidence Interval (C.I.) from linear mixed effect regression models, with scattered radiation measurements (451P) (μGy) as the dependent variable and Field of View (FOV) as the independent variable, also adjusting for height of measurements ^1^ made and coordinates (*x*, *y*) of measurement points, by room.

Room (CBCT)	FOV	β (μGy)	95% C.I. for β (μGy)	*p*-Value
1	4 × 4	Reference category		
	6 × 6	0.480	(0.139 to 0.820)	0.006 *
	8 × 8	0.709	(0.365 to 1.053)	<0.001 **
2	8 × 8	Reference category		
	11 × 8	−0.257	(−0.805 to 0.291)	0.358
3	8 × 8	Reference category		
	15 × 15	2.005	(1.453 to 2.558)	<0.001 **

^1^ height of measurements varies only in rooms 1 and 2. Room (CBCT): X-ray room during cone-beam computed tomography (CBCT). FOV: Field of View. C.I.: Confidence Interval; β: beta coefficient * statistically significant, α = 5% ** statistically significant, α = 1‰.

## Data Availability

The datasets used and/or analyzed during the current study are available from the corresponding author upon reasonable request.

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
