# Peer review of "Scattered Radiation Distribution Utilizing Three Different Cone-Beam Computed Tomography Devices for Maxillofacial Diagnostics: A Research Study"

_jcm, 2023, doi:10.3390/jcm12196199_

Round 1

Reviewer 1 Report

Dear Editor,

I would like to thank you the opportunity to review the manuscript entitled “Scattered radiation distribution utilizing three different cone beam computed tomography devices” and so contribute with this Journal.

The present study aimed to estimate scattered radiation and its spatial distribution around CBCT devices. Although more than 270 publications on the subject have been identified (PubMed), the present study offers an interesting contribution. Minor modifications are required for the manuscript to be indicated for publication.

See below for my suggestions

Introduction

  • Second paragraph: "The x-ray beam of the CBCT machine consists of primary radiation that yields useful imaging information through the patient and secondary radiation which is scattered radiation.” Authors must insert references that support the assertions made.
  • Second paragraph: "The primary radiation is produced within the x-ray tube, enters the patient, interacts with human tissues and attenuates variably in the area under examination, conveying the useful information about the structures to be imaged.” Authors must insert references that support the assertions made.
  • Page 3, line 6: Replace [8,9,10] by [8-10].
  • The introduction consists of long paragraphs. However, some aspects were not contemplated. They are: Is the study original? If so, what makes this study original? What has already been studied on the subject? What gap does the study intend to fill?
  • Would it be possible to present a null hypothesis?

Materials and Methods

  • Study material: Would it be possible to create a table with all exposure parameters? This type of presentation can help the reader to understand the different groups evaluated.
  • Add the number of measurements taken on the third CBCT device.
  • Present the heights of the measurements according to the rooms.
  • Why did the number of measurements vary between the three devices?
  • Would it be possible to show the dimensions of the rooms?

Results

  • Table 1, 2, 3 and 4: Authors should review table captions. Some abbreviations were not contemplated.

Discussion

  • If the authors agree with the inclusion of a null hypothesis, this could be answered in the first paragraph.
  • What are the advantages and disadvantages of the method used in the study?
  • What are the future prospects opened by this study?

Conclusion

-  The conclusion must be redone. This should be limited to responding to the research objectives.

Author Response

Response to Reviewer 1 Comments

1. Summary

2. Questions for General Evaluation

Reviewer’s Evaluation

Response and Revisions

Does the introduction provide sufficient background and include all relevant references?

Must be improved

We have, accordingly, modified the Introduction section.

Are all the cited references relevant to the research?

Yes

Thank you for your evaluation

Is the research design appropriate?

Can be improved

We have, accordingly, modified the Materials and Methods section.

Are the methods adequately described?

Can be improved

We have, accordingly, modified the Materials and Methods section.

Are the results clearly presented?

Yes

Thank you very much for your evaluation.

Are the conclusions supported by the results?

Can be improved

We have, accordingly, modified the Conclusions section.

3. Point-by-point response to Comments and Suggestions for Authors

Comment 0: The present study aimed to estimate scattered radiation and its spatial distribution around CBCT devices. Although more than 270 publications on the subject have been identified (PubMed), the present study offers an interesting contribution. Minor modifications are required for the manuscript to be indicated for publication.

Response 0: We thank you very much for your positive opinion.

·         Comments 1: [Introduction]: Second paragraph: "The x-ray beam of the CBCT machine consists of primary radiation that yields useful imaging information through the patient and secondary radiation which is scattered radiation.” Authors must insert references that support the assertions made.

Response 1: Thank you for pointing this out. We agree with this comment. Following your suggestion we have insert the appropriate references [Introduction Section; Paragraph 2; Page 2]: “The x-ray beam of the CBCT machine consists of primary radiation that yields useful imaging information through the patient and secondary radiation which is scattered radiation [1].

·         Comments 2: [Introduction]: Second paragraph: "The primary radiation is produced within the x-ray tube, enters the patient, interacts with human tissues and attenuates variably in the area under examination, conveying the useful information about the structures to be imaged.” Authors must insert references that support the assertions made.

Response 2: Thank you again for your comment. Following your suggestion we have insert the appropriate references [Introduction Section; Paragraph 2; Page 2, 3]: “The primary radiation is produced within the x-ray tube, enters the patient, interacts with human tissues and attenuates variably in the area under examination, conveying the useful information about the structures to be imaged [1, 2].”

Comments 3: [Introduction]: Page 3, line 6: Replace [8,9,10] by [8-10].

Response 3: Thank you for this comment. We have now modified the text accordingly [Introduction Section; Paragraph 2; Page 3, line 6]: “Thus, scattered radiation has a negative effect on image quality [7] and essentially stands as the main factor to contribute in reduced spatial resolution, reduced contrast resolution, and increased noise in CBCT [8-10].”

·         Comments 4: [Introduction]: The introduction consists of long paragraphs. However, some aspects were not contemplated. They are: Is the study original? If so, what makes this study original? What has already been studied on the subject? What gap does the study intend to fill?

Response 4: We thank you so much for this insightful comment. We agree with your above consideration that although more than 270 publications on the subject have been identified (PubMed), the present study offers an interesting contribution. This original research study aims to estimate the patterns of scattered radiation and its spatial distribution around three CBCT devices for maxillo-facial diagnostics. Following your suggestion we have incorporated that information in the Title and Introduction Section [Title, Page 1]: Scattered radiation distribution utilizing three different cone beam computed tomography devices for maxillo-facial diagnostics: a research study”; [Introduction Section; Paragraph 5; Page 3]: “This original research study specifically aims: to estimate the patterns of scattered radiation and its spatial distribution around three CBCT devices, in order to determine potential positions for an operator exceptionally to stand if needed to be present in the x-ray room during the CBCT examination for maxillo-facial diagnostics.”

Comments 5: [Introduction]: Would it be possible to present a null hypothesis?

Response 5: Thank you for this suggestion. A null hypothesis is a type of statistical hypothesis that proposes that no statistical significance exists in a set of given observations. “Hypothesis testing” is used to assess the credibility of a hypothesis by using sample data. In our study, we used “hypothesis testing”. This was already presented in Introduction Section [Introduction Section; Paragraph 4; Page 3]: Our hypothesis assumes that in some exceptional occasions the dentist or the staff (dental assistant, radiology technologist) may need to be present in the x-ray room during the CBCT examination, to visually and verbally direct and encourage the patient to a correct placement. Thus, we wanted to determine if there is a safety distance from the CBCT device to stand, so to receive the lowest possible scattered radiation [6, 11].”

·         Comments 6: [Materials and Methods]: Study material: Would it be possible to create a table with all exposure parameters? This type of presentation can help the reader to understand the different groups evaluated.

Response 6: Thank you so much for this suggestion. We have now modified the text according your suggestion in Materials and Methods Section [Materials and Methods Section; 2.1. Study material; Paragraph 1; Page 4]: “…Exposure measurements were performed for different kVp, mA and Field of View (FOV) values [(CBCT1; exposure time: 9sec, kVp: 90kV, mA: 7mA, voxel size: 0.125mm), (CBCT2; exposure time: 3.6sec, kVp: 90kV, mA: 3.66mA, voxel size: 0.125mm), (CBCT3; exposure time: 3.6sec, kVp: 110kV, mA: 3.66mA, voxel size: 0.3mm)]…” . Measurements of scattered radiation from different Field of Views (FOVs) were also performed. This was already presented in Materials and Methods Section and in Table 2, Table 4 [[Materials and Methods Section; 2.1. Study material; Paragraph 5; Page 6]: “Measurements of scattered radiation from different Field of Views (FOVs) were also performed. More specifically, measurements of scattered radiation were carried out in the CBCT1 device in three different FOVs [(4 × 4), (6 × 6), (8 × 8)], in the CBCT2 device in two different FOVs [(8 × 8), (11 × 8)] and in the CBCT3 device in two different FOVs [(8 × 8), (15 × 15)]…”. Table 2 presents the distribution of 451P measurements, for different FOV in each room. Table 4 presents the results from linear mixed effect regression models, with 451P measurement as the dependent variable and FOV as the independent variable, also adjusting for height of measurements made (only for room 1 and 2) and coordinates (x, y) of measurement points, in each room. These were already presented in Results Section [Results Section; Paragraph 2, Page 8]: “Table 2 presents the distribution of 451P measurements, for different FOV in each room.”; [Results Section; Paragraph 4, Page 10]: “Table 4 presents the results from linear mixed effect regression models, with 451P measurement as the dependent variable and FOV as the independent variable, also adjusting for height of measurements made (only for room 1 and 2) and coordinates (x, y) of measurement points, in each room.”

·         Comments 7: [Materials and Methods]: Add the number of measurements taken on the third CBCT device.

Response 7: Thank you for this comment. The number of measurements taken on the third CBCT device (36 measurements) was already presented in Materials and Methods Section and in Table 1 [Materials and Methods Section; 2.1. Study material; Paragraph 4; Page 6]: “…in the CBCT3 room all measurements (36 measurements) were carried out…”.

·         Comments 8: [Materials and Methods]: Present the heights of the measurements according to the rooms.

Response 8: Thank you for your suggestion. Nevertheless, this was already presented in Materials and Methods Section and in Table 1 [Materials and Methods Section; 2.1. Study material; Paragraph 4; Page 6]: “191 (CBCT1) and 32 (CBCT2) measurements of scattered radiation at two different heights (1m or 1.3m from the floor) were carried out in the rooms of CBCT1 and CBCT2 devices at each point of intersection of the x and y axes (yellow points) (Fig 2). The measurement at a height of 1m from the floor, represented the anatomical location of the gonads as well as at a height of 1.3m from the floor, represented the anatomical location of the thyroid gland [21]. It is of importance that in the CBCT3 room all measurements (36 measurements) were carried out at the same height (1.3m), due to technical difficulties (room and device restrictions).”

·         Comments 9: [Materials and Methods]: Why did the number of measurements vary between the three devices?

Response 9: Thank you so much for this comment. Indeed, the complex floor plan of rooms 2 and 3 did not allow additional measurements in these areas. So, 191 measurements were carried out in CBCT1 room, while 32 and 36 measurements were carried out in CBCT 2 and CBCT3 room, respectively.

Comments 10: [Materials and Methods]: Would it be possible to show the dimensions of the rooms?

Response 10: Thank you for this comment. The dimensions of the rooms were already presented in Figure 3 (room 1: 2.8×3m), Figure 4 (room 2: 1.5×2m) and Figure 5 (room 3: 2×2.6m). In order to make this clearer to the readers we show now the dimensions of the rooms in the Results Section, according your suggestion [Figure Legends 3, 4, 5; Page 11, 12, 13]: “Figure 3. Color map of spatial distribution of scattered radiation (μGy) in room 1 (2.8×3m)”, “Figure 4. Color map of spatial distribution of scattered radiation (μGy) in room 2 (1.5×2m)”, “Figure 5. Color map of spatial distribution of scattered radiation (μGy) in room 3 (2×2.6m)”.

·         Comments 11: [Results]: Table 1, 2, 3 and 4: Authors should review table captions. Some abbreviations were not contemplated.

Response 11: Thank you so much for this insightful comment. Following your suggestion we have now modify Table Captions (Table 1, 2, 3, 4) and we have now add all the appropriate abbreviations [Results Section; Pages 7-11]:

Table 1; Pages 7-8:

Room (CBCT): x-ray room during Cone Beam Computed Tomography (CBCT).

Points: The points represented the fixed positions of the Victoreen ionization chamber (Inovision 451P) at the same distances from the CBCT device; measurements (n): the number of measurements of scattered radiation which was carried out in the rooms (CBCT1, CBCT2, CBCT3).

Height (m): The measurement at a height of 1m from the floor represented the anatomical location of the gonads as well as at a height of 1.3m from the floor, represented the anatomical location of the thyroid gland.

Scattered radiation 451P measurements (μGy): Scattered radiation measurements were taken with a Victoreen ionization chamber (Inovision 451P).

Mean: mean value of scattered radiation (451P) (μGy) measurements.

SD: Standard Deviation of scattered radiation (451P) (μGy) measurements.

Maximum: maximum value of scattered radiation (451P) (μGy) measurements.

Distance from point “0”: Distance (cm) from CBCT device. Point “0” represented the fixed position of the CBCT device in each room.

Table 2; Page 9:

Room (CBCT): x-ray room during Cone Beam Computed Tomography (CBCT).

FOV: Field of View; SD: Standard Deviation.

Scattered radiation 451P measurements (μGy): Scattered radiation measurements were taken with a Victoreen ionization chamber (Inovision 451P).

Mean: mean value of scattered radiation (451P) (μGy) measurements.

SD: Standard Deviation of scattered radiation (451P) (μGy) measurements.

1One-way analysis of variance (ANOVA)

2independent samples t-test

*statistically significant, α=5%

**statistically significant, α=1%ο

Table 3; Page 10:

1height of measurements varies only in rooms 1 and 2

Room (CBCT): x-ray room during Cone Beam Computed Tomography (CBCT).

Distance (m): 1-meter increase in the distance (m) from CBCT device.

C.I.: Confidence Interval; β: beta coefficient

*statistically significant, α=5%

**statistically significant, α=1%ο

Table 4; Page 10-11:

1height of measurements varies only in rooms 1 and 2

Room (CBCT): x-ray room during Cone Beam Computed Tomography (CBCT).

FOV: Field of View.

C.I.: Confidence Interval; β: beta coefficient

*statistically significant, α=5%

**statistically significant, α=1%ο

·         Comments 12: [Discussion]: If the authors agree with the inclusion of a null hypothesis, this could be answered in the first paragraph.

Response 12: Thank you again for this comment. As we mentioned above (Response 5), in the present study, we used “hypothesis testing”. This was already presented in Introduction Section [Introduction Section; Paragraph 4; Page 3]: “Our hypothesis assumes that in some exceptional occasions the dentist or the staff (dental assistant, radiology technologist) may need to be present in the x-ray room during the CBCT examination. Thus, we wanted to determine if there is a safety distance from the CBCT device to stand, so to receive the lowest possible scattered radiation [6, 11].”

·         Comments 13: [Discussion]: What are the advantages and disadvantages of the method used in the study?

Response 13: We thank you very much for this comment. A disadvantage of the present study was that we did not use angle measurements when placing the ionization chamber (451P) in the three rooms with the CBCT devices. This was already presented in Discussion Section [Discussion Section; Paragraph 3; Page 15]: “…A limitation of our study was that we did not use angle measurements when placing the ionization chamber (451P) in the three rooms with the CBCT devices…”. Another disadvantage of our study was that in the room 3 (CBCT3) all measurements were made at the same height (1.3m). This was already presented in Discussion Section [Discussion Section; Paragraph 5; Page 16]: “…In addition, due to technical difficulties, a limitation of the present study was that in the room 3 (CBCT3) all measurements were made at the same height (1.3m)…”. Nevertheless, a comparative advantage of our study was that the measurement of the scattered radiation was carried out in three different CBCT devices and numerous measurements of the scattered radiation were carried out at two different heights. This was already presented in Discussion Section [Discussion Section; Paragraph 5; Page 16]: “…A comparative advantage of our study was that the measurement of the scattered radiation was carried out in three different CBCT devices and numerous measurements of the scattered radiation were carried out at two different heights…”.

·         Comments 14: [Discussion]: What are the future prospects opened by this study?

Response 14: Thank you so much for your suggestion. In this study we used three CBCT devices in order to estimate scattered radiation and its spatial distribution. The present study opens future prospects for further investigation of scattered radiation distribution with more CBCT devices for maxillo-facial diagnostics. Following your suggestion we have now incorporated that information in the Discussion Section [Discussion Section; Paragraph 5; Page 16]: “The present study opens future prospects for further investigation of scattered radiation distribution with more CBCT devices for maxillo-facial diagnostics.”

Comments 15: [Conclusion]: The conclusion must be redone. This should be limited to responding to the research objectives.

Response 15: Thank you for this comment. We have now modified the Conclusions Section in the revised manuscript [Conclusions Section]: “In all CBCT devices that were tested in this study the scattered radiation that an individual may be exposed to, is significantly decreased with distance. Especially, the Newtom VGi CBCT showed the greatest decrease with distance. In all instances the measured scattered radiation was below 1mGy, which is defined by the radiation protection guidelines as the exposure radiation limit of the general population. Nevertheless, the scattered radiation is significantly reduced, as long as the dentist or radiology technologist or other occupationally exposed individual exceptionally stand at a safe distance and position from the patient (as determined by our results) in the x-ray room during the CBCT examination for maxillo-facial diagnostics.”

4. Response to Comments on the Quality of English Language

Point 1: English language fine. No issues detected.

Response 1: Thank you so much for your consideration.

5. Additional clarifications

We have no further clarifications.

Reviewer 2 Report

Overall, the study focusses on the essential issue of radio protection. This also affects dentistry and maxillofacial surgery. Above all the use of CBCT is increasing. Authors presented data of different devices and rooms and showed a significant reduction of scatter radiation, at least in two rooms. As a consequence, they argue that it is safe for personal to stay in the room. This should be described more in detail and relativized.

The correct template is missing. Line number is missing and makes an easy review more difficult.

Title: 

Please clarify that the issue is about Maxillo-facial diagnostics.

Kind of study should be clarified in the title.

Abstract:

Consider stochastic radiation damage – “relatively safe” should not be mentioned in combination with radiation protection.

“Therefore, at certain distances from the central x-ray, the scattered radiation was below the critical 1mGy. Consequently, an operator could stay inside the room accompanying the patient being scanned, if necessary.” à Please clarify why 1mGy should be the critical limit. Conclusion to justify personal inside the room should not be the aim. It is in contrast to radiation protection. It seems to justify staying inside for each investigation. Please change.

Introduction:

“Low-dose dental cone-beam computed tomography (CBCT) is one of the most important technological achievements in oral and maxillofacial radiology in the last forty years.” Please clarify: What do you mean with Low-dose? The mode instead of normal or HD? If you mean that CBCT is generally low dose compared to CT, this is wrong.

„Our hypothesis assumes that in some exceptional occasions the dentist or the staff (dental assistant, radiology technologist) may need to be present in the x-ray room during the CBCT examination. Thus, we wanted to determine if there is a safety distance from the CBCT device to stand, so to receive the lowest possible scattered radiation [6, 11]. “Please clarify the reasons why anybody should stay into the x-ray room? It might by for fixation of the patient. In this case increased distance doesn’t help.

“relatively safe positions for an operator to stand if needed to be present in the x-ray room during the CBCT examination.” Don’t mention this for reason of the study. Better might be a reduction of strict shielding of the room.

A description of the kind of the study is missing.

Study material:

„The following devices were tested in this research study:“à Add the range of FOV these devices offer.

“Scattered radiation measurements were performed at the same distances from the CBCT devices (Fig 2).” How often measurements were repeated per mode and device?

191 (CBCT1) and 32 (CBCT2) measurements of scattered radiation at two different heights (1m or 1.3m from the floor) were carried out in the rooms of CBCT1 and CBCT2 devices at each point of intersection of the x and y axes (yellow points) (Fig 2).”

Information’s about the FOV and exposure parameters, mode, adult/child, voxle size, exposure time… are missing, Were measurements repeated or just performed once per FOV;This could bias results. A table might be helpful. 

Were results of both heights summed up? Were there differences between heights.

Description of the room dimension/ shielding, position of door is missing. Might have influence.

Results

Table 1: As already mentioned: Detailed information about the exposure parameters is missing. Were there repetitions of measurements at each point.

Why there was only 1.3m measurement at device 3?

SD:à (+/-) should be added.

Table 1: Scattered radiation 451P measurements (μGy) column is misleading. 

A uniformity of table design should be used, also for table 1.

Figure 3,4: A different scaling makes a comparison difficult.

Discussion:

of the people who by exception could be present inside the x-ray room were determined (>100cm from CBCT1, >50cm from CBCT2 and >55cm from CBCT3).” à for which reason?

Please clarify if your results make radio protection rooms with shielding of the walls unnecessary, even there are clear legal regulations.

To argue and underline a reduction of wall shielding instead of the possibility of staff in the room is more sensible.

Please discuss the limitation of just a hand full FOV you tested. What about larger field of views? Could your conclusion be generalized or are there more measurements necessary. Have legal regulations be altered?

Author Response

Response to Reviewer 2 Comments

1. Summary

2. Questions for General Evaluation

Reviewer’s Evaluation

Response and Revisions

Does the introduction provide sufficient background and include all relevant references?

Must be improved

We have, accordingly, modified the Introduction section.

Are all the cited references relevant to the research?

Yes

Thank you for your evaluation

Is the research design appropriate?

Can be improved

We have, accordingly, modified the Materials and Methods section.

Are the methods adequately described?

Must be improved

We have, accordingly, modified the Materials and Methods section.

Are the results clearly presented?

Can be improved

We have, accordingly, modified the Results section.

Are the conclusions supported by the results?

Must be improved

We have, accordingly, modified the Conclusions section.

3. Point-by-point response to Comments and Suggestions for Authors

Comments 0: Overall, the study focusses on the essential issue of radio protection. This also affects dentistry and maxillofacial surgery. Above all the use of CBCT is increasing. Authors presented data of different devices and rooms and showed a significant reduction of scatter radiation, at least in two rooms. As a consequence, they argue that it is safe for personal to stay in the room. This should be described more in detail and relativized. The correct template is missing. Line number is missing and makes an easy review more difficult.

Response 0: Thank you so much for your positive consideration. We apologize for the incorrect template. Perhaps an error occurred during the editing process.

Comments 1: [Title]: Please clarify that the issue is about Maxillo-facial diagnostics. Kind of study should be clarified in the title.

Response 1: Thank you for pointing this out. We agree with this comment. Following your suggestion we have now incorporated that information in the Title [Title, Page 1]: Scattered radiation distribution utilizing three different cone beam computed tomography devices for maxillo-facial diagnostics: a research study”.

Comments 2: [Abstract]: Consider stochastic radiation damage – “relatively safe” should not be mentioned in combination with radiation protection.

Response 2: Thank you for this comment. We have, accordingly, modified the revised text to emphasize this point in the Abstract Section [Abstract, Page 1]: “This study aimed to estimate scattered radiation and its spatial distribution around three Cone Beam Computed Tomography (CBCT) devices, in order to determine potential positions for an operator exceptionally to stand if needed to be inside the CBCT room.”

Comments 3: [Abstract]: “Therefore, at certain distances from the central x-ray, the scattered radiation was below the critical 1mGy. Consequently, an operator could stay inside the room accompanying the patient being scanned, if necessary. .” à Please clarify why 1mGy should be the critical limit. Conclusion to justify personal inside the room should not be the aim. It is in contrast to radiation protection. It seems to justify staying inside for each investigation. Please change.

Response 3: We thank you so much for this insightful comment. We have now modified the revised text, according your suggestion [Abstract, Page 2]: “Therefore, at certain distances from the central x-ray, the scattered radiation was below the critical 1mGy, which is defined by the radiation protection guidelines as the exposure radiation limit of the general population. Consequently, an operator could exceptionally stay inside the room accompanying the patient being scanned, if necessary.”

Comments 4: [Introduction]: “Low-dose dental cone-beam computed tomography (CBCT) is one of the most important technological achievements in oral and maxillofacial radiology in the last forty years.” Please clarify: What do you mean with Low-dose? The mode instead of normal or HD? If you mean that CBCT is generally low dose compared to CT, this is wrong.

Response 4: Thank you very much for this comment. We have now modified the revised text, according your consideration in Introduction Section [Introduction Section, Paragraph 1, Page 2]: “Cone-beam computed tomography (CBCT) is one of the most important technological achievements in oral and maxillofacial radiology in the last forty years.”

Comments 5: [Introduction]: “Our hypothesis assumes that in some exceptional occasions the dentist or the staff (dental assistant, radiology technologist) may need to be present in the x-ray room during the CBCT examination. Thus, we wanted to determine if there is a safety distance from the CBCT device to stand, so to receive the lowest possible scattered radiation [6, 11]”. Please clarify the reasons why anybody should stay into the x-ray room? It might by for fixation of the patient. In this case increased distance doesn’t help.

Response 5: Thank you so much for this comment. We have now revised the text, according your suggestion [Introduction Section, Paragraph 4, Page 3]: “Our hypothesis assumes that in some exceptional occasions the dentist or the staff (dental assistant, radiology technologist) may need to be present in the x-ray room during the CBCT examination, to visually and verbally direct and encourage the patient to a correct placement. Thus, we wanted to determine if there is a safety distance from the CBCT device to stand, so to receive the lowest possible scattered radiation [6, 11].”

Comments 6: [Introduction]: “relatively safe positions for an operator to stand if needed to be present in the x-ray room during the CBCT examination.” Don’t mention this for reason of the study. Better might be a reduction of strict shielding of the room.

Response 6: Thank you for this comment. We have now modified the text, in order to be clearer to the readers [Introduction Section, Paragraph 5, Page 3]: “This original research study specifically aims: to estimate the patterns of scattered radiation and its spatial distribution around three CBCT devices, in order to determine potential positions for an operator exceptionally to stand if needed to be present in the x-ray room during the CBCT examination for maxillo-facial diagnostics.”

Comments 7: [Introduction]: A description of the kind of the study is missing.

Response 7: We thank you for this consideration. We have now added the kind of the study in Introduction Section, according your suggestion [Introduction Section, Paragraph 5, Page 3]: “This original research study specifically aims: to estimate the patterns of scattered radiation and its spatial distribution around three CBCT devices, in order to determine potential positions for an operator exceptionally to stand if needed to be present in the x-ray room during the CBCT examination for maxillo-facial diagnostics.”

Comments 8: [Study material]: “The following devices were tested in this research study”: à Add the range of FOV these devices offer.

Response 8: Thank you so much for this comment. This was already presented in Materials and Methods Section [Materials and Methods Section; 2.1. Study material; Paragraph 5, Page 6]: “Measurements of scattered radiation from different Field of Views (FOVs) were also performed. More specifically, measurements of scattered radiation were carried out in the CBCT1 device in three different FOVs [(4 × 4), (6 × 6), (8 × 8)], in the CBCT2 device in two different FOVs [(8 × 8), (11 × 8)] and in the CBCT3 device in two different FOVs [(8 × 8), (15 × 15)].”

Comments 9: [Study material]: “Scattered radiation measurements were performed at the same distances from the CBCT devices (Fig 2).” How often measurements were repeated per mode and device?

Response 9: Thank you so much for this comment. Measurements were repeated 3 times per mode and device. However, the measurement results were repeatable and identical for each measurement. Therefore, this was not mentioned in the text to avoid confusing the readers.

Comments 10: [Study material]: “191 (CBCT1) and 32 (CBCT2) measurements of scattered radiation at two different heights (1m or 1.3m from the floor) were carried out in the rooms of CBCT1 and CBCT2 devices at each point of intersection of the x and y axes (yellow points) (Fig 2).” Information’s about the FOV and exposure parameters, mode, adult/child, voxle size, exposure time… are missing, Were measurements repeated or just performed once per FOV;This could bias results. A table might be helpful. 

Response 10: Thank you very much for this suggestion. We have now modified the text according your suggestion in Materials and Methods Section [Materials and Methods Section; 2.1. Study material; Paragraph 1; Page 4]: “…Exposure measurements were performed for different kVp, mA and Field of View (FOV) values [(CBCT1; exposure time: 9sec, kVp: 90kV, mA: 7mA, voxel size: 0.125mm), (CBCT2; exposure time: 3.6sec, kVp: 90kV, mA: 3.66mA, voxel size: 0.125mm), (CBCT3; exposure time: 3.6sec, kVp: 110kV, mA: 3.66mA, voxel size: 0.3mm)]…” . Measurements of scattered radiation from different Field of Views (FOVs) were also performed. This was already presented in Materials and Methods Section and in Table 2, Table 4 [[Materials and Methods Section; 2.1. Study material; Paragraph 5; Page 6]: “Measurements of scattered radiation from different Field of Views (FOVs) were also performed. More specifically, measurements of scattered radiation were carried out in the CBCT1 device in three different FOVs [(4 × 4), (6 × 6), (8 × 8)], in the CBCT2 device in two different FOVs [(8 × 8), (11 × 8)] and in the CBCT3 device in two different FOVs [(8 × 8), (15 × 15)]…”. Table 2 presents the distribution of 451P measurements, for different FOV in each room. Table 4 presents the results from linear mixed effect regression models, with 451P measurement as the dependent variable and FOV as the independent variable, also adjusting for height of measurements made (only for room 1 and 2) and coordinates (x, y) of measurement points, in each room. These were already presented in Results Section [Results Section; Paragraph 2, Page 8]: “Table 2 presents the distribution of 451P measurements, for different FOV in each room.”; [Results Section; Paragraph 4, Page 10]: “Table 4 presents the results from linear mixed effect regression models, with 451P measurement as the dependent variable and FOV as the independent variable, also adjusting for height of measurements made (only for room 1 and 2) and coordinates (x, y) of measurement points, in each room.”

Comments 11: [Study material]: Were results of both heights summed up? Were there differences between heights.

Response 11: Thank you for this comment. All statistical models were adjusted for the height (m) that the measurement was carried out and for the coordinates (x, y) of each measurement point, by including a bivariate smooth function (thin plate spline) of (x, y). This was already presented in Materials and Methods Section [Materials and Methods Section; 2.2. Statistical Analysis; Paragraph 1, Pages 6 and 7]: “…All models were adjusted for the height (m) that the measurement was carried out and for the coordinates (x, y) of each measurement point, by including a bivariate smooth function (thin plate spline) of (x, y)…”. Furthermore, there were not statistically significant differences between heights. This was already presented in Results Section [Results Section; Paragraph 3; Page 9]: “… Note that in both rooms, the height of the measurements did not significantly predict 451P (p = 0.956 and p = 0.323, respectively)…”; [Results Section; Paragraph 5; Page 11]: “…Measurement height did not statistically significantly differentiate the measurement of scattered radiation in the ionization chamber.”

Comments 12: [Study material]: Description of the room dimension/ shielding, position of door is missing. Might have influence.

Response 12: Thank so much for this comment. The room dimensions were already presented in Figure 3 (room 1: 2.8×3m), Figure 4 (room 2: 1.5×2m) and Figure 5 (room 3: 2×2.6m). In order to make this clearer to the readers we show now the dimensions of the rooms in the Results Section, according your suggestion [Figure Legends 3, 4, 5; Page 11, 12, 13]: “Figure 3. Color map of spatial distribution of scattered radiation (μGy) in room 1 (2.8×3m)”, “Figure 4. Color map of spatial distribution of scattered radiation (μGy) in room 2 (1.5×2m)”, “Figure 5. Color map of spatial distribution of scattered radiation (μGy) in room 3 (2×2.6m)”.

Comments 13: [Results]: Table 1: As already mentioned: Detailed information about the exposure parameters is missing. Were there repetitions of measurements at each point.

Response 13: Thank you again for this comment. As we mentioned above (Response 9), measurements were repeated 3 times per mode and device. However, the measurement results were repeatable and identical for each measurement. Therefore, this was not mentioned in the text to avoid confusing the readers. Also, as we mentioned above in Response 10, the exposure parameters are now presented in Materials and Methods Section [Materials and Methods Section; 2.1. Study material; Paragraph 1; Page 4]: “…Exposure measurements were performed for different kVp, mA and Field of View (FOV) values [(CBCT1; exposure time: 9sec, kVp: 90kV, mA: 7mA, voxel size: 0.125mm), (CBCT2; exposure time: 3.6sec, kVp: 90kV, mA: 3.66mA, voxel size: 0.125mm), (CBCT3; exposure time: 3.6sec, kVp: 110kV, mA: 3.66mA, voxel size: 0.3mm)]…”.

Comments 14: [Results]: Why there was only 1.3m measurement at device 3?

Response 14: Thank you so much for this comment. In the CBCT3 room all measurements (36 measurements) were carried out at the same height (1.3m), due to technical difficulties (room and device restrictions). This was already presented in Materials and Methods Section [Materials and Methods Section; 2.1. Study Material; Paragraph 4; Page 6]: “…It is of importance that in the CBCT3 room all measurements (36 measurements) were carried out at the same height (1.3m), due to technical difficulties (room and device restrictions).”

Comments 15: [Results]: SD:à (+/-) should be added.

Response 15: Thank you for this comment. We have now added SD (±) in Table 1, according your suggestion [Results Section; Pages 7]:

Table 1; Pages 7-8:

Room (CBCT): x-ray room during Cone Beam Computed Tomography (CBCT).

Points: The points represented the fixed positions of the Victoreen ionization chamber (Inovision 451P) at the same distances from the CBCT device; measurements (n): the number of measurements of scattered radiation which was carried out in the rooms (CBCT1, CBCT2, CBCT3).

Height (m): The measurement at a height of 1m from the floor represented the anatomical location of the gonads as well as at a height of 1.3m from the floor, represented the anatomical location of the thyroid gland.

Scattered radiation 451P measurements (μGy): Scattered radiation measurements were taken with a Victoreen ionization chamber (Inovision 451P).

Mean: mean value of scattered radiation (451P) (μGy) measurements.

SD: Standard Deviation of scattered radiation (451P) (μGy) measurements.

Maximum: maximum value of scattered radiation (451P) (μGy) measurements.

Distance from point “0”: Distance (cm) from CBCT device. Point “0” represented the fixed position of the CBCT device in each room.

Comments 16: [Results]: Table 1: Scattered radiation 451P measurements (μGy) column is misleading. 

Response 16: Thank you so much for this helpful comment. Following your suggestion we have now modify all Table Captions (Table 1, 2, 3, 4) and we have now add all the appropriate abbreviations [Results Section; Pages 7-11]:

Table 1; Pages 7-8:

Room (CBCT): x-ray room during Cone Beam Computed Tomography (CBCT).

Points: The points represented the fixed positions of the Victoreen ionization chamber (Inovision 451P) at the same distances from the CBCT device; measurements (n): the number of measurements of scattered radiation which was carried out in the rooms (CBCT1, CBCT2, CBCT3).

Height (m): The measurement at a height of 1m from the floor represented the anatomical location of the gonads as well as at a height of 1.3m from the floor, represented the anatomical location of the thyroid gland.

Scattered radiation 451P measurements (μGy): Scattered radiation measurements were taken with a Victoreen ionization chamber (Inovision 451P).

Mean: mean value of scattered radiation (451P) (μGy) measurements.

SD: Standard Deviation of scattered radiation (451P) (μGy) measurements.

Maximum: maximum value of scattered radiation (451P) (μGy) measurements.

Distance from point “0”: Distance (cm) from CBCT device. Point “0” represented the fixed position of the CBCT device in each room.

Table 2; Page 9:

Room (CBCT): x-ray room during Cone Beam Computed Tomography (CBCT).

FOV: Field of View; SD: Standard Deviation.

Scattered radiation 451P measurements (μGy): Scattered radiation measurements were taken with a Victoreen ionization chamber (Inovision 451P).

Mean: mean value of scattered radiation (451P) (μGy) measurements.

SD: Standard Deviation of scattered radiation (451P) (μGy) measurements.

1One-way analysis of variance (ANOVA)

2independent samples t-test

*statistically significant, α=5%

**statistically significant, α=1%ο

Table 3; Page 10:

1height of measurements varies only in rooms 1 and 2

Room (CBCT): x-ray room during Cone Beam Computed Tomography (CBCT).

Distance (m): 1-meter increase in the distance (m) from CBCT device.

C.I.: Confidence Interval; β: beta coefficient

*statistically significant, α=5%

**statistically significant, α=1%ο

Table 4; Page 10-11:

1height of measurements varies only in rooms 1 and 2

Room (CBCT): x-ray room during Cone Beam Computed Tomography (CBCT).

FOV: Field of View.

C.I.: Confidence Interval; β: beta coefficient

*statistically significant, α=5%

**statistically significant, α=1%ο

Comments 17: [Results]: A uniformity of table design should be used, also for table 1.

Response 17: Thank you again for this comment. As we mentioned above (Response 16) we have now modify all Table Captions (Table 1, 2, 3, 4) and we have now add all the appropriate abbreviations.

Comments 18: [Results]: Figure 3,4: A different scaling makes a comparison difficult.

Response 18: Thank you for this comment. We agree with your consideration. However, we cannot interfere with the color map creation software to improve the scaling of the image.

Comments 19: [Discussion]: “of the people who by exception could be present inside the x-ray room were determined (>100cm from CBCT1, >50cm from CBCT2 and >55cm from CBCT3).” à for which reason?

Response 19: Thank you for this comment. As we mentioned above (Response 5), in some exceptional occasions the dentist or the staff (dental assistant, radiology technologist) may need to be present in the x-ray room during the CBCT examination, to visually and verbally direct and encourage the patient to a correct placement.

Comments 20: [Discussion]: Please clarify if your results make radio protection rooms with shielding of the walls unnecessary, even there are clear legal regulations.

Response 20: Thank you so much for this insightful comment. In the present study it was observed that scattered radiation decreases as we move away from the x-ray beam. However, this by no means implies that our results make radio protection rooms with shielding of the walls unnecessary. This is now presented in Discussion Section [Discussion Section; Paragraph 4; Page 16]: “…There were in agreement with the results of previous studies. However, this by no means implies that our results make radio protection rooms with shielding of the walls unnecessary.”

Comments 21: [Discussion]: To argue and underline a reduction of wall shielding instead of the possibility of staff in the room is more sensible.

Response 21: Thank you again for this comment. Please see above for our responses to this comment (Responses 5, 6 and 20).

Comments 22: [Discussion]: Please discuss the limitation of just a hand full FOV you tested. What about larger field of views? Could your conclusion be generalized or are there more measurements necessary. Have legal regulations be altered?

Response 22: Thank you very much for this comment. This study did not aim to alter legal regulations. In this study the following devices were tested: Morita Accuitomo (CBCT1), Newtom Giano HR (CBCT2), Newtom VGi (CBCT3). Measurements of scattered radiation were carried out in the CBCT1 device in three different FOVs [(4 × 4), (6 × 6), (8 × 8)], in the CBCT2 device in two different FOVs [(8 × 8), (11 × 8)] and in the CBCT3 device in two different FOVs [(8 × 8), (15 × 15)]. There was no larger FOV on these devices. So, the present study opens future prospects for further investigation of scattered radiation distribution with more CBCT devices for maxillo-facial diagnostics. Following your suggestion we have now incorporated that information in the Discussion Section [Discussion Section; Paragraph 5; Page 16]: “…The present study opens future prospects for further investigation of scattered radiation distribution with more CBCT devices for maxillo-facial diagnostics.”

4. Response to Comments on the Quality of English Language

Point 1: I am not qualified to assess the quality of English in this paper

Response 1: Thank you again for your time and effort to review this manuscript

5. Additional clarifications

We have no further clarifications.

Round 2

Reviewer 2 Report

Thank you for your modifications. From my point there are no further corrections needed.